# Investigating the coupling relationships of railway safety risks using the *N-K* model and complex network theory

Jiaxu Chen[1,2], Lin Zhao[1,2]*, Jinghui Liu[1,2], Gaolei Wang[1,2], Zhan Guo[1,2]

**1** China Academy of Railway Sciences, Railway Science & Technology Research & Development Center, Beijing, China, **2** Railway Safety Research Center, China State Railway Group Co., Ltd., Beijing, China

* 1522617580@qq.com

## Abstract

To quantitatively analyze the coupling relationships between railway safety risk factors, identify key factors contributing to railway accidents, and develop scientific strategies for accident prevention, this study introduces a complex network-based *N-K* model to investigate the coupling relationships of railway safety risk factors. First, we identified 18 railway safety risk factors by analyzing case data from railway accidents. The occurrence probabilities and coupling values of these risk factors were then calculated using the N-K model. Subsequently, based on the constructed railway safety risk complex network, reachability and centrality analyses were performed to determine the key factors of railway safety risk. Results indicate that the occurrence of railway accidents is directly proportional to the risk coupling value; the greater the number of coupling factors, the higher the risk value. The coupling of personnel factors and equipment factors is particularly prone to leading to railway accidents. Conversely, effective management of the coupling between personnel and equipment factors can significantly reduce the likelihood of accidents. Inadequate maintenance and unsafe human behavior were identified as critical factors contributing to railway accidents and should be prioritized in prevention efforts.

## Introduction

The railway system is a typical complex system, characterized by a complex organizational structure and diverse operational scenarios. It is highly susceptible to both external and internal interference events, which can significantly increase safety risks. During railway system operations, the mutual coupling among multiple risk factors can exacerbate these risks, potentially leading to railway accidents [1]. Therefore, analyzing railway safety risk factors and studying their coupling relationships are crucial for preventing and controlling railway accidents. These analyses are essential for enhancing railway safety risk management and reducing accident incidence rates, thereby playing a significant role in the development of the railway industry.

**Data availability statement:** The data utilized in this study are the property of the Railway Safety Research Center, China State Railway Group Co., Ltd. Researchers interested in obtaining accident and fault data should contact the Railway Safety Research Center directly via telephone ((86) 010-51871181). The authors do not possess any special access rights or privileges. All other data necessary for the study are fully included within the paper.

**Funding:** The author(s) disclosed the receipt of the following financial support for the research, authorship, and/or publication of this article: This work was supported by the Science and Technology Research and Development Program of China National Railway Corporation Limited (Grant No. K2024T002).

**Competing interests:** The authors have declared that no competing interests exist.

## Literature review

Currently, numerous scholars have conducted research in the field of railway safety risk, employing various methods such as Failure Mode and Effects Analysis (FMEA), Fault Tree Analysis (FTA), Event Tree Analysis (ETA), Bayesian Networks (BN), and Complex Networks (CN) [ 2–7]. However, existing research has predominantly focused on the dynamic changes of individual risks, with relatively limited exploration of the coupling relationships among risk factors.

**1) Dynamic risk assessment model.** Wang et al. [8] proposed a prediction system based on the SCR-Gaussian DI method, using tensor analysis to conduct dynamic risk assessments in railways. Peng et al. [9] extended the traditional Fault Tree Analysis method to time events and fault characteristics, constructing a timed fault tree model to evaluate the risks of railway transportation systems. Xu et al. [10] proposed a composite risk analysis method based on Bayesian networks and bow-tie models to predict the accident risks of systems. Liu Yang et al. [11] proposed a Text-based Bayesian Network (TBN) method to establish a Bayesian Network (BN) based on text records for dynamic analysis of railway derailment risks.

**2) Development of an accident prediction and risk mitigation model.** Liu et al. [12] introduced a hazard prediction method based on knowledge graphs to capture and prevent potential accident risk pathways, thereby preventing railway operation accidents. Liu et al. [12] employed a railway accident prediction method based on knowledge graphs to describe the multiple potential characteristics of risks and block the causal pathways of accidents. Guo et al. [13] improved the similarity aggregation method combined with fuzzy Bayesian networks to analyze the changing processes in accident risk assessment.

**3) Railway hazardous materials transportation risk analysis model.** Ebrahimi et al. [14] analyzed the uncertain risk factors and their consequences in railway hazardous materials transportation accidents through case studies. Hosseini et al. [15] proposed a risk assessment method based on risk value to measure the risks in the process of railway hazardous materials transportation. Zarei et al. [16] developed a dynamic risk analysis model for hazardous materials railway transportation based on dynamic Bayesian networks, studying the domino effect risks under uncertain conditions.

## Motivation and contributions

This paper analyzes railway safety risk coupling relationships using railway accident cases. Risk coupling is defined as the comprehensive risk arising from the interaction of two or more risk factors. We innovatively apply the N-K model to calculate coupling values between railway safety risk factors and perform quantitative coupling risk analysis. The coupling value, a quantitative indicator derived from the N-K model, reflects the intensity of the collaborative effect of multiple risk factors and their non-linear interactive risks. Using complex network theory, we establish a key network of railway safety risk factors. Risk reachability and network node centrality analyses are then employed to investigate the mechanisms of these risk factors. Both analyses are

indicators in complex networks that measure the importance of nodes (risk factors) in transmission paths. The coupling values calculated by the N-K model are corrected to identify the final key risk factors. Our findings provide a decision-making basis for railway safety risk prevention and control. The goal is to implement effective measures at the source of accidents to enhance railway safety management.

## Analysis of risk factors and coupling relationships in railway safety

### Analysis of risk factors in railway safety management

The railway system comprises numerous subsystems, including civil engineering, rolling stock, and various other components. It is characterized by its complexity, intricate relationships among components, and diverse operational environments [17]. The variability of these environments, the diversity of its structural composition, and the complexity of its interconnections render the railway system susceptible to interference from both internal and external factors. Adverse weather conditions, construction activities near the railway lines, or internal system failures can disrupt the safe and stable operation of the railway, potentially leading to accidents that result in casualties and significant economic losses. To conduct an in-depth analysis of the safety risk factors associated with railway accidents, this study compiles and reviews data from railway accidents both domestically and internationally. Through literature research and data collection, a total of 157 accident cases were gathered. Statistical analysis was performed from the perspectives of both internal railway system components and external environmental factors contributing to the accidents. The results are presented in Fig 1.

In the statistical analysis of railway system accidents, equipment and facility failures account for the highest number of incidents. These failures primarily include contact line malfunctions, ATP system failures, and turnout malfunctions. Additionally, accident factors encompass unsafe behaviors of external personnel, errors by internal staff, and inadequate on-site management. Drawing on systems engineering and safety management theories [18], accidents are not isolated

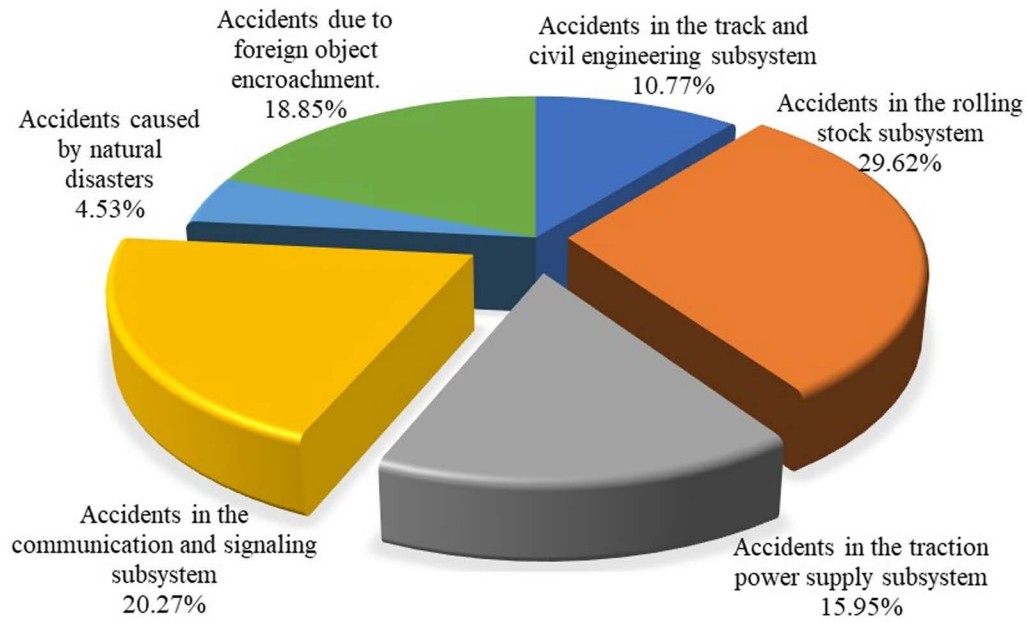

**Fig 1. Statistical map of railway accident distribution.**

events but rather the outcomes of the interplay and mutual influence among four categories of factors: personnel, equipment, environment, and management, as depicted in Fig 2.

During the operation of the railway system, the couplings among personnel, equipment, environment, and management are particularly significant. Railway transportation involves numerous interconnected components and complex systems, and any deviation in one factor can trigger a chain reaction through the interdependencies among these factors, ultimately leading to accidents. Through an in-depth analysis of railway accidents, safety risk factors have been systematically categorized into four main groups: personnel, equipment, environment, and management. These categories are further subdivided into a total of 18 specific risk factors, as detailed in Fig 3.

## Investigating the coupling relationships of railway safety risk factors

In physics, coupling is defined as the mutual influence and interaction between two or more elements. In complex systems, numerous components and intricate structures exist, with elements being interrelated and affecting one another. Consequently, the coupling relationships within these systems are highly complex. A change in one element can impact others, potentially leading to systemic change. Risk coupling refers to the phenomenon where, during normal system operation, various interrelated risk factors interact continuously, influencing system stability. The impact and interdependence of these risk factors determine the degree of coupling. As the number of risk couplings increases, so does the coupling degree, leading to greater system instability and continuous risk accumulation. This, in turn, raises the likelihood of accidents. To deeply investigate the relationships among risk factors in railway accidents, this study conducts a coupling evolution analysis of railway accident risks from four perspectives: personnel, equipment, environment, and management. The interaction mechanisms among these four types of risk factors are qualitatively examined. Fig 4 illustrates the coupled evolution mechanism of railway safety risk.

When the railway system operates normally, a single risk occurrence can trigger a risk coupling event. The railway system's control measures (i.e., protective measures) immediately respond to mitigate the risk, thereby increasing the system's risk level. If control measures are effectively implemented or the influence of objective factors is reduced, the risk factors are contained, the system's protective layer remains intact, and the system continues to function normally,

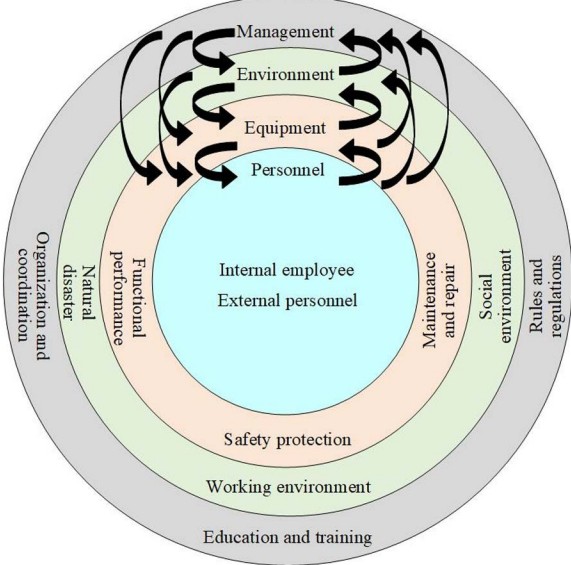

**Fig 2. Schematic diagram of the influence of accident factors.**

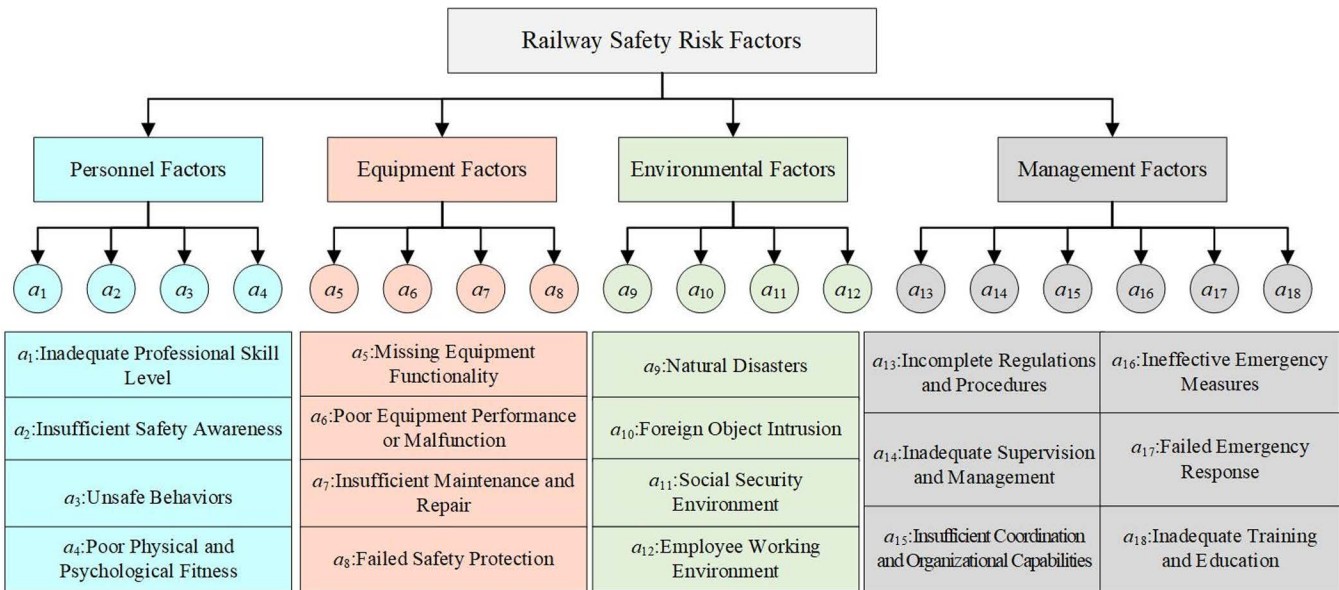

**Fig 3. Railway safety risk factors.**

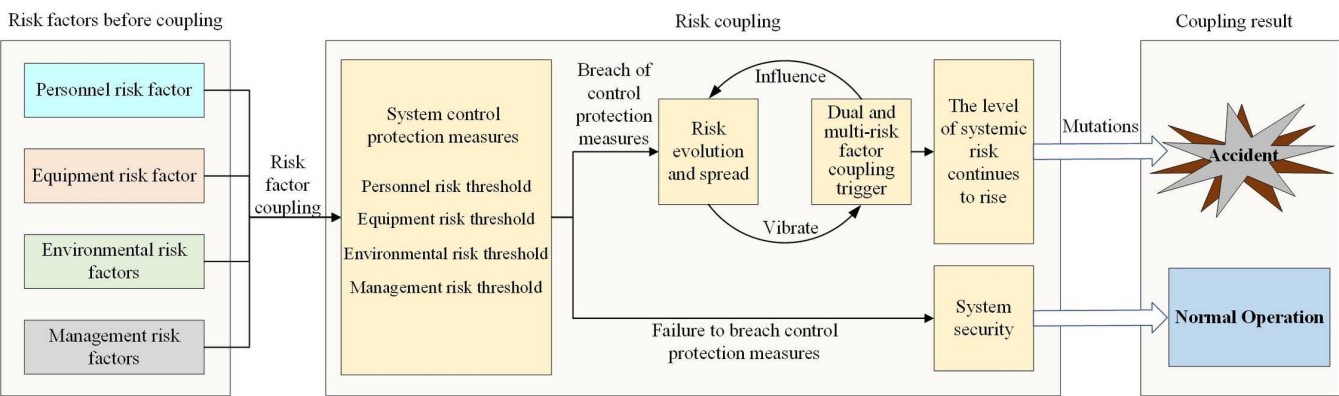

**Fig 4. Coupled evolution mechanism of railway safety risk.**

preventing a railway accident. Conversely, if control measures fail or the influence of objective factors intensifies, the risk level escalates. This escalation can cause risk propagation and coupling, destabilizing the system and leading to accidents. The types of railway safety risk coupling are illustrated in Fig 5.

## Railway safety risk assessment via complex network analysis using the *N-K* model

### Construction of the N-K risk coupling model for railway systems

The *N-K* model is a classical tool for coupling analysis of complex systems [19–21]. Its theoretical construction is based on two core variables: the number of elements $N$ in the system and the degree of interaction between elements $K$. By quantifying the nonlinear couplings between system elements, the *N-K* model provides a theoretical framework for

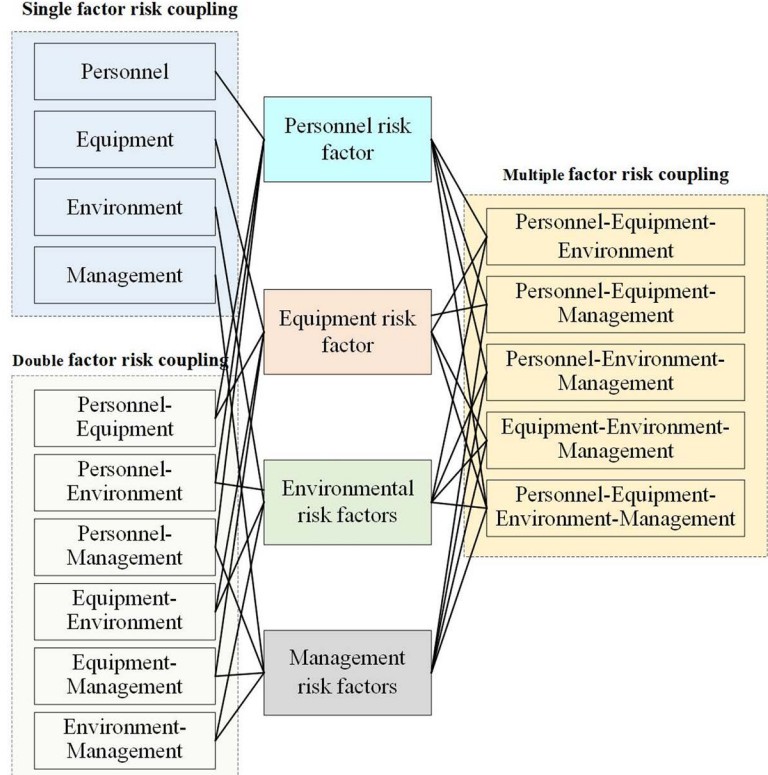

**Fig 5. Risk coupling type diagram.**

analyzing multidimensional risk coupling mechanisms. Here, $N$ represents the number of components in the system, and its value directly influences the dimensionality of the system state combination space. When the system comprises $N$ elements, each with $n$ possible states, the system state combination space exhibits exponential growth, with the maximum number of combined states being $n^N$. The variable $K$ reflects the strength of coupling correlations between elements, with its value ranging from 0 to $N-1$. The dynamic variation of $K$ directly determines the system's complexity. Specifically, when $K=0$, the system is in a completely decoupled state, with each element evolving independently. When $K=1$, the system exhibits linear correlation characteristics. As $K$ increases to $K\geq2$, typical nonlinear coupling features emerge, and couplings between elements produce significant coupling effects. In this study, the $N$-$K$ model is applied to the coupling analysis of railway safety risk. The coupling interaction information $T$ of risk factors, including personnel, equipment, environment, and management, is calculated to quantify the impact of risk factor coupling on railway accidents. The process is as follows:

**Step 1: Calculation of the coupling probability $p$ for risk factors.** The risk factor coupling probability $p$ encompasses the coupling probabilities of single, double, and multiple risk factors, and the calculation formula is as follows:

$$p_{h,i,j,k} = m_{h,i,j,k}/c \tag{1}$$

Where, $p_{h,i,j,k}$ represents the probability of risk coupling occurring when the personnel risk factor is in state $h$, the equipment risk factor is in state $i$, the environmental risk factor is in state $j$, and the management risk factor is in state $k$, $m_{h,i,j,k}$

denotes the number of occurrences of different types of railway accidents, $c$ represents the total number of railway accidents.

**Step 2: Coupling probability $p....$ associated with specific risk factors.** When a risk factor occurs, $p_{...}$ is expressed as the coupling probability $p$ related to the occurrence of that risk factor. For example, when the equipment factor occurs, the risk probability of the factor changing is denoted as $p_{.i..}$ The formula is as follows:

$$p_{.i..} = p_{.1..} = p_{0100} + p_{1100} + p_{0110} + p_{0101} + p_{1110} + p_{1101} + p_{0111} + p_{1111} \tag{2}$$

Therefore, the risk probability $p$ of single, double, and multiple risk factors under different states can be calculated.

**Step 3: Coupled interactive information T calculation.** In the *N-K* model of railway safety risk, the calculation formula for multi-risk coupling considering four factors is as follows:

$$T_4 = (a, b, c, d) = \sum_{h=1}^{H}\sum_{i=1}^{I}\sum_{j=1}^{J}\sum_{k=1}^{K} p_{h,i,j,k} \times \log_2[p_{h,i,j,k}/(p_{h...} \times p_{.i..} \times p_{..j.} \times p_{...k})] \tag{3}$$

Where, $a$ represents the personnel factor, $b$ represents the equipment factor, $c$ represents the environmental factor, $d$ represents the management factor. $h$ denotes the state of the personnel factor, where $h = 0,1,2,…, H$ (with $H$ being the number of types of personnel factors), $i$ denotes the state of the equipment factor, where $i = 0,1,2,…, I$ (with $I$ being the number of types of equipment factors), $j$ denotes the state of the environmental factor, where $j = 0,1,2,…, J$ (with $J$ being the number of types of environmental factors), $k$ denotes the state of the management factor, where $k = 0, 1, 2,…, K$ (with $K$ being the number of types of management factors). $p_{h,i,j,k}$ is the probability of the coupling of the four risk factors $a$, $b$, $c$, and $d$ occurring in states $h$, $i$, $j$, and $k$, respectively. $p_{h...}$ is the probability of factor $a$ occurring in state $h$, $p_{.i..}$ is the probability of factor $b$ occurring in state $i$, $p_{..j.}$ is the probability of factor $c$ occurring in state $j$, $p_{...k}$ is the probability of factor $d$ occurring in state $k$. $T_4$ ($a$, $b$, $c$, $d$) represents the coupling value of the four risk factors $a$, $b$, $c$, and $d$ in states $h$, $i$, $j$, and $k$, respectively. A higher coupling value indicates a greater likelihood of the coupling of these risk factors occurring, and consequently, a higher probability of an accident occurring.

When considering the multi-risk coupling among three factors, four distinct couplings are identified: personnel-equipment-environment, personnel-equipment-management, personnel-environment-management, and equipment-environment-management. These couplings are respectively denoted as $T_{31}$ ($a$, $b$, $c$), $T_{32}$ ($a$, $b$, $d$), $T_{33}$ ($a$, $c$, $d$), and $T_{34}$ ($b$, $c$, $d$). The calculation formula for these couplings is as follows:

$$T_3 = \begin{cases} T_{31}(a, b, c) = \sum_{h=1}^{H}\sum_{i=1}^{I}\sum_{j=1}^{J} p_{h,i,j} \times \log_2[p_{h,i,j}/(p_{h...} \times p_{.i..} \times p_{..j.})] \\ T_{32}(a, b, d) = \sum_{h=1}^{H}\sum_{i=1}^{I}\sum_{k=1}^{K} p_{h,i,k} \times \log_2[p_{h,i,k}/(p_{h...} \times p_{.i..} \times p_{...k})] \\ T_{33}(a, c, d) = \sum_{h=1}^{H}\sum_{j=1}^{J}\sum_{k=1}^{K} p_{h,j,k} \times \log_2[p_{h,j,k}/(p_{h...} \times p_{..j.} \times p_{...k})] \\ T_{34}(b, c, d) = \sum_{i=1}^{I}\sum_{j=1}^{J}\sum_{k=1}^{K} p_{i,j,k} \times \log_2[p_{i,j,k}/(p_{.i..} \times p_{..j.} \times p_{...k})] \end{cases} \tag{4}$$

When considering the coupling of dual risk factors, six distinct couplings are identified: personnel-equipment, personnel-environment, personnel-management, equipment-environment, equipment -management, and environment-management. These couplings are denoted as $T_{21}$ ($a$,$b$), $T_{22}$ ($a$, $c$), $T_{23}$ ($a$, $d$), $T_{24}$ ($b$, $c$), $T_{25}$ ($b$, $d$), and $T_{26}$ ($c$, $d$), respectively. The calculation formula for these couplings is as follows:

$$T_2 = \begin{cases} T_{21}(a,b) = \sum_{h=1}^{H}\sum_{i=1}^{I} p_{h,i} \times \log_2[p_{h,i}/(p_{h...} \times p_{.i..})] \\ T_{22}(a,c) = \sum_{h=1}^{H}\sum_{j=1}^{J} p_{h,j} \times \log_2[p_{h,j}/(p_{h...} \times p_{..j.})] \\ T_{23}(a,d) = \sum_{h=1}^{H}\sum_{k=1}^{K} p_{h,k} \times \log_2[p_{h,k}/(p_{h...} \times p_{...k})] \\ T_{24}(b,c) = \sum_{i=1}^{I}\sum_{j=1}^{J} p_{i,j} \times \log_2[p_{i,j}/(p_{.i..} \times p_{..j.})] \\ T_{25}(b,d) = \sum_{i=1}^{I}\sum_{k=1}^{K} p_{i,k} \times \log_2[p_{i,k}/(p_{.i..} \times p_{...k})] \\ T_{26}(c,d) = \sum_{j=1}^{J}\sum_{k=1}^{K} p_{j,k} \times \log_2[p_{j,k}/(p_{..j.} \times p_{...k})] \end{cases}$$

(5)

### Railway safety risk factor analysis based on the N-K model and complex network theory

The interweaving and mutual influence of railway safety risk factors make it difficult for traditional risk analysis methods to provide comprehensive and accurate assessments. Moreover, risk communication exhibits distinct network characteristics. Therefore, leveraging the theory of complex networks, we model risk factors as nodes and the communication process as edges within the network [22–24]. By constructing a railway safety risk complex network, we clarify the roles and functions of different risks in the transmission process. Integrating the unique algorithm of the *N-K* model, we conduct an in-depth analysis of the coupling relationships between different risk factors and refine the risk nodes within the complex network. This approach enables us to accurately identify the most critical risk factors affecting the stability of the complex network, thereby providing a more targeted and efficient basis for decision-making in railway safety risk management and effectively ensuring the safe and stable operation of the railway transportation system. The analysis method for railway safety risk factors based on the *N-K* model and complex network is illustrated in Fig 6.

To address the characteristic parameters in complex networks, two key indicators—closeness centrality and betweenness centrality—are selected for the analysis of railway safety risk. Closeness centrality reflects the extent to which a node acts as a bridge along the shortest paths between other nodes in the network. The greater the value of this index, the more critical the node is in connecting different parts of the network. It can be expressed as the ratio of the number of shortest paths passing through the node to the total number of shortest paths between all pairs of nodes. Betweenness centrality, on the other hand, measures the proximity of a node to all other nodes in the network. The greater the value of this index, the higher the proximity of a node to other nodes in the network. It can be expressed as the reciprocal of the sum of the shortest paths between the node and all other nodes in the network. The calculation formula for closeness centrality is as follows:

$$C_c(i) = \left[\sum_{j=1}^{n} d(i,j)\right]^{-1} \qquad i \neq j$$

(6)

Where, $C_c(i)$ denotes the closeness centrality of node $i$; $n$ is the total number of nodes in the network; and $d(i,j)$ represents the shortest path distance between nodes $i$ and $j$.

Betweenness centrality reflects the extent to which a node in the network influences overall propagation. A higher value of this indicator indicates that the node appears in multiple critical transmission paths within the network, thereby exerting a greater influence on the overall propagation of the network. The calculation formula is as follows:

$$C_B(i) = \sum_{j}^{n}\sum_{k}^{n} b_{jk}(i) = \sum_{j}^{n}\sum_{k}^{n} \frac{g_{jk}(i)}{g_{jk}}$$
$$j \neq k \neq i, j < k$$

(7)

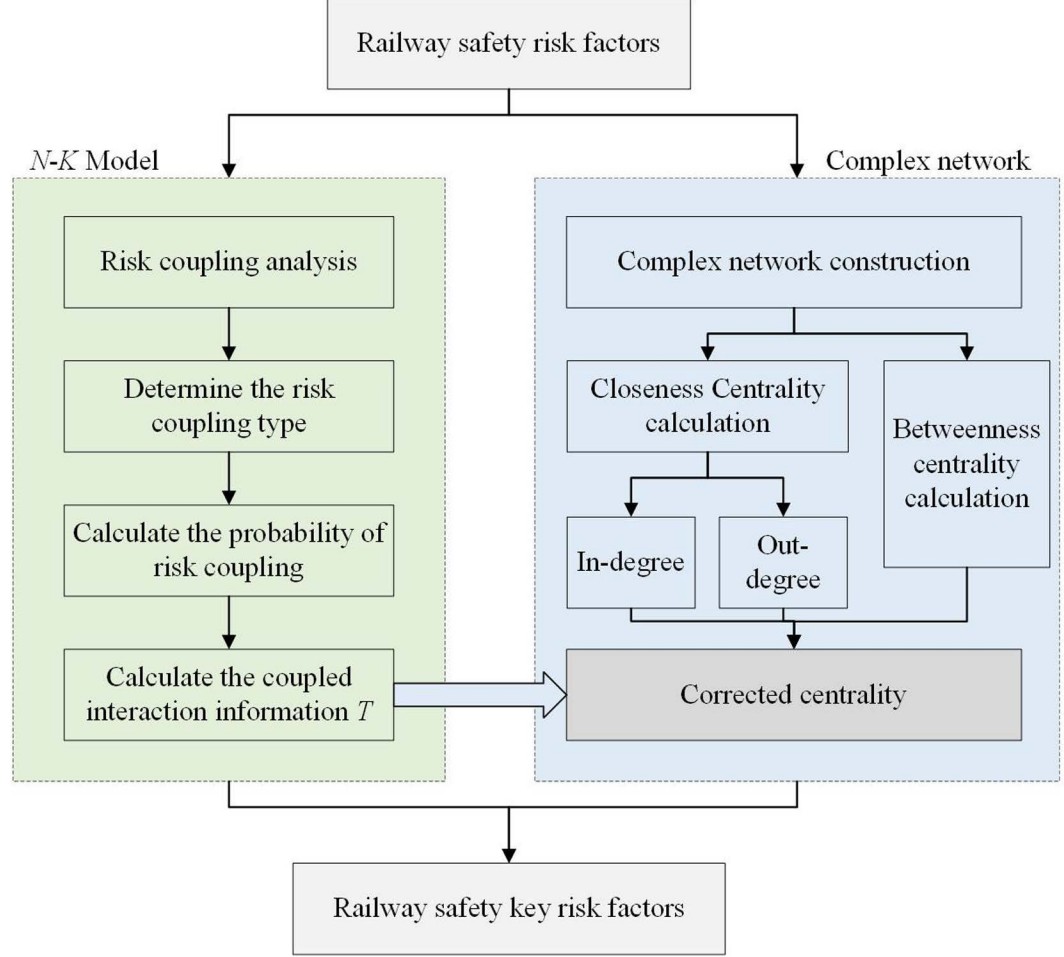

**Fig 6. Complex network railway safety risk factor analysis method based on N-K model.**

Where, $C_B(i)$ denotes the betweenness centrality of node $i$; $n$ represents the total number of nodes in the network. $b_{jk}(i)$ is the probability that node $i$ lies on the shortest path between nodes $j$ and $k$; $g_{jk}(i)$ is the number of shortest paths between nodes $j$ and $k$ that pass through node $i$; $g_{jk}(i)$ is the total number of shortest paths between nodes $j$ and $k$.

To elucidate the meaning of the selected indicators rigorously, this study formulates a complex-network model in which nodes denote elementary system entities and edges denote their latent interactions. Building on this representation, the nodal closeness and betweenness centralities are computed analytically via Equations (6) and (7), respectively.

Among the computed centrality metrics, node $x_9$ attains the highest closeness centrality, signifying its minimal average geodesic distance to all other nodes in the network. Concurrently, node $x_{10}$ records the greatest betweenness centrality, functioning as the single most critical articulation point for inter-node traffic. Fig 7 visualizes these findings: red-marked $x_9$ exhibits a markedly shorter mean shortest-path length, indicating superior global accessibility; conversely, blue-marked $x_{10}$ participates in the largest fraction of shortest paths, establishing itself as the primary structural pivot. However, this pivotal role also renders $x_{10}$ a potential congestion bottleneck under high-demand scenarios.

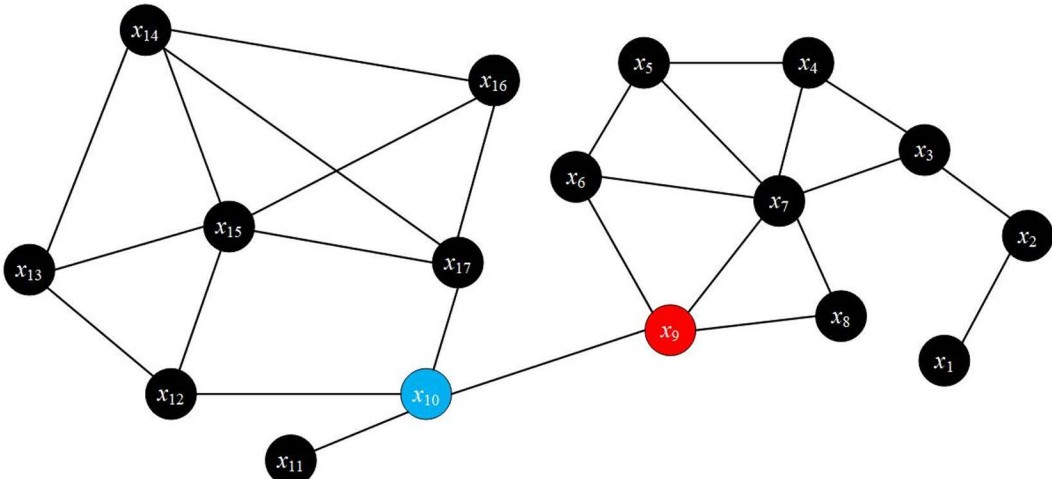

**Fig 7. Diagram illustrating the concept of closeness centrality and betweenness centrality in complex networks.**

## Case study analysis of railway safety risks using the N-K model and complex network theory

### Analysis of N-K model calculation results

Based on the 157 railway accidents collected, a detailed analysis of the specific factors contributing to accident causes was conducted, and the occurrence frequencies of four categories of risks—personnel, equipment, environment, and management—were tallied. Railway safety risk factors were represented by two states: "0" indicates that the risk did not occur, while "1" indicates that the risk contributed to a railway accident. Using statistical analysis of the 157 railway accident reports, the number and probability of single, double, and multi-factor couplings formed by personnel, equipment, environment, and management factors were calculated and analyzed. The calculation results are presented in Table 1.

**Table 1. Coupling analysis of railway safety risk factors.**

| Coupling type | Coupling factors | Coupling frequency | Coupling probability |
|---|---|---|---|
| Single Factor | Uniformly uncoupled | 0 | $P_{0000}=0$ |
| | Personnel | 16 | $P_{1000}=0.1019$ |
| | Equipment | 55 | $P_{0100}=0.3503$ |
| | Environment | 10 | $P_{0010}=0.0637$ |
| | Management | 8 | $P_{0001}=0.0510$ |
| Double Factors | Personnel-Equipment | 9 | $P_{1100}=0.0573$ |
| | Personnel-Environment | 4 | $P_{1010}=0.0255$ |
| | Personnel-Management | 10 | $P_{1001}=0.0637$ |
| | Equipment-Environment | 6 | $P_{0110}=0.0382$ |
| | Equipment-Management | 8 | $P_{0101}=0.0510$ |
| | Environment-Management | 5 | $P_{0011}=0.0318$ |
| Multiple Factors | Personnel-Equipment-Environment | 6 | $P_{1110}=0.0382$ |
| | Personnel-Equipment-Management | 7 | $P_{1101}=0.0446$ |
| | Personnel-Environment-Management | 5 | $P_{1011}=0.0318$ |
| | Equipment-Environment-Management | 7 | $P_{0111}=0.0446$ |
| | Personnel-Equipment-Environment-Management | 1 | $P_{1111}=0.0064$ |

1) Calculation of Single-Factor Risk Coupling Probabilities: When assessing single-factor risk coupling, it is necessary to consider the coupling risks associated with personnel, equipment, environment, and management. For instance, the probability of personnel risk factors being involved in the coupling can be calculated as $P_{1...} = P_{1000} + P_{1100} + P_{1010} + P_{1001} + P_{1110} + P_{1011} + P_{1101} + P_{1111}$, Consequently, the probability value for each single risk factor can be determined, with the results presented in Table 2.

2) Calculation of Probabilities for Different Forms of Dual-Factor Risk Coupling: Under various conditions of dual-factor risk coupling, there are combinations of personnel-equipment, personnel-environment, personnel-management, equipment-environment, and environment-management couplings. For instance, the probability that both personnel and equipment factors are involved in the risk coupling is calculated as $P_{11..} = P_{1100} + P_{1110} + P_{1101} + P_{1111}$. Consequently, the probability values for dual risks can be determined, with the results presented in Table 3.

3) Calculation of Probabilities for Various Forms of Multi-Factor Risk Coupling. In scenarios involving multi-factor risk coupling, there exist five distinct combinations of risk coupling forms: personnel-device-environment, personnel-device-management, personnel-environment-management, device-environment-management, and personnel-device-environment-management. For instance, the probability that personnel, equipment, and environmental factors simultaneously engage in risk coupling is denoted as $P_{111.} = P_{1110} + P_{1111}$. Thus, the probability values for multiple risks can be ascertained, with the results detailed in Table 4.

4) Calculation of Railway Safety Risk Coupling Values. Based on the identified risk factors, formulas (3)–(5) are employed to calculate the coupling risk values for both two-factor and multi-factor couplings within the railway safety risk framework. The resultant calculations are presented in Table 5.

**Table 2. Probabilities of risk factor couplings.**

| Coupling form | $P_{0...}$ | $P_{1...}$ | $P_{.0..}$ | $P_{.1..}$ |
|---|---|---|---|---|
| Probability | 0.3822 | 0.4194 | 0.3694 | 0.6206 |
| Coupling form | $P_{..0.}$ | $P_{..1.}$ | $P_{...0}$ | $P_{...1}$ |
| Probability | 0.7198 | 0.2802 | 0.6754 | 0.3294 |

**Table 3. Probability of occurrence of double-factor risk coupling form.**

| Coupling form | $P_{00..}$ | $P_{01..}$ | $P_{10..}$ | $P_{11..}$ | $P_{0.0.}$ | $P_{0.1.}$ | $P_{1.0.}$ | $P_{1.1.}$ |
|---|---|---|---|---|---|---|---|---|
| Probability | 0.1465 | 0.4841 | 0.2229 | 0.1465 | 0.4523 | 0.1783 | 0.2675 | 0.1019 |
| Coupling form | $P_{0..0}$ | $P_{0..1}$ | $P_{1..0}$ | $P_{1..1}$ | $P_{.00.}$ | $P_{.01.}$ | $P_{.10.}$ | $P_{.11.}$ |
| Probability | 0.4522 | 0.1784 | 0.1784 | 0.1465 | 0.2166 | 0.1528 | 0.4932 | 0.1274 |
| Coupling form | $P_{.0.0}$ | $P_{.0.1}$ | $P_{.1.0}$ | $P_{.1.1}$ | $P_{..00}$ | $P_{..01}$ | $P_{..10}$ | $P_{..11}$ |
| Probability | 0.1911 | 0.1783 | 0.4840 | 0.1466 | 0.5095 | 0.2103 | 0.1656 | 0.1146 |

**Table 4. Probability of occurrence of multi-factor risk coupling form.**

| Coupling form | $P_{000.}$ | $P_{001.}$ | $P_{010.}$ | $P_{011.}$ | $P_{100.}$ | $P_{110.}$ | $P_{101.}$ | $P_{111.}$ |
|---|---|---|---|---|---|---|---|---|
| Probability | 0.0318 | 0.0955 | 0.4013 | 0.0828 | 0.1656 | 0.1019 | 0.0573 | 0.0446 |
| Coupling form | $P_{00.0}$ | $P_{00.1}$ | $P_{01.0}$ | $P_{01.1}$ | $P_{10.0}$ | $P_{11.0}$ | $P_{10.1}$ | $P_{11.1}$ |
| Probability | 0.0637 | 0.0828 | 0.3885 | 0.0956 | 0.1274 | 0.0955 | 0.0955 | 0.0510 |
| Coupling form | $P_{0.00}$ | $P_{0.10}$ | $P_{0.01}$ | $P_{0.11}$ | $P_{1.00}$ | $P_{1.10}$ | $P_{1.01}$ | $P_{1.11}$ |
| Probability | 0.3503 | 0.1019 | 0.0956 | 0.0764 | 0.1592 | 0.0637 | 0.1083 | 0.0382 |
| Coupling form | $P_{.000}$ | $P_{.010}$ | $P_{.001}$ | $P_{.011}$ | $P_{.100}$ | $P_{.110}$ | $P_{.101}$ | $P_{.111}$ |
| Probability | 0.1019 | 0.0892 | 0.1147 | 0.0636 | 0.4076 | 0.0764 | 0.0956 | 0.0510 |

**Table 5. Risk coupling results and ranking.**

| Value of coupling | Rank |
|---|---|
| $T_4$ (a, b, c, d) =0.756 | 1 |
| $T_{31}$ (a, b, c) =0.468 | 4 |
| $T_{32}$ (a, b, d) =0.561 | 2 |
| $T_{33}$ (a, c, d) =0.396 | 5 |
| $T_{34}$ (b, c, d) =0.116 | 8 |
| $T_{21}$ (a, b) =0.503 | 3 |
| $T_{22}$ (a, c) =0.388 | 6 |
| $T_{23}$ (a, d) =0.355 | 7 |
| $T_{24}$ (b, c) =0.014 | 10 |
| $T_{25}$ (b, d) =0.024 | 9 |
| $T_{26}$ (c, d) =0.012 | 11 |

As shown in Table 5, the risk coupling values are ranked as $T_4$ (a, b, c, d) > $T_{32}$ (a, b, d) > $T_{21}$ (a, d) > $T_{31}$ (a, b, c) > $T_{33}$ (a, c, d) > $T_{22}$ (a, c) > $T_{23}$ (a, d) > $T_{34}$ (b, c, d) > $T_{25}$ (b, d) > $T_{24}$ (b, c) > $T_{26}$ (c, d). This ranking indicates that railway safety risk increases with the increase in both the type and quantity of coupling factors.

The coupling results of each factor are analyzed in conjunction with the calculation results of the *N-K* model and the coupling interaction combination type. The coupling value of railway safety risk factors is directly proportional to the number of risk factors involved in the coupling. Specifically, the coupling values follow the order $T_4 > T_3 > T_2 > T_1$. This trend indicates that as the number of risk factors involved in coupling increases, the coupling risk correspondingly rises, thereby increasing the probability of accidents. In the analysis of two-factor risk coupling, the combinations $T_{21}$ (a, b)、$T_{22}$ (a, c) 、$T_{23}$ (a, d) exhibit relatively large risk coupling values. Notably, all these combinations include personnel risk factors, suggesting that personnel factors readily couple with other risk factors. Among these, $T_{21}$ (a, b) stands out with a significantly higher coupling risk value compared to the other two-factor combinations, indicating that the coupling of personnel risk and equipment risk is particularly likely to contribute to railway safety accidents in the context of two-factor coupling risks. Among the three-factor risk coupling combinations, $T_{31}$ (a, b, c) and $T_{32}$ (a, b, d) display larger risk coupling values. A common characteristic of these combinations is the inclusion of both personnel and equipment factors, further emphasizing that these factors are more likely to couple with other risk factors. A comprehensive analysis reveals that, on the basis of the existing coupling between personnel and equipment factors, the addition of environmental or management factors significantly increases the likelihood of railway safety accidents. This finding underscores the substantial threat posed to railway operational safety by such multi-factor couplings.

**Construction and analysis of the complex network of railway safety risk factors**

**(1) Risk centrality analysis.** By collecting a large number of railway accident cases and analyzing their causes, we have summarized the adjacency matrix of risk factors affecting railway safety,The value rule for the elements in the adjacency matrix (Table 6) is as follows: If factor *i* points to factor *j* (i.e., *i* is the antecedent triggering factor of *j*), then the adjacency matrix $G_{ij} = 1$ (indicating a directed edge $i \rightarrow j$). Conversely, if there is no record or consensus of "*i* pointing to *j*," then $G_{ij} = 0$. Based on this matrix, a complex network diagram of railway safety risk factors has been constructed, as shown in Fig 8.

In the complex network of railway safety risk factors, we calculated the risk nodes, proximity centrality, and intermediate centrality. Given that the network is directed, proximity centrality comprises two metrics: out-degree and in-degree. The out-degree reflects how readily a risk factor can induce other risk factors, while the in-degree indicates the extent to which a risk factor is influenced by other risks. The centrality values of the risk factors are presented in Table 7. As shown in

**Table 6. Adjacency matrix.**

| | $a_1$ | $a_2$ | $a_3$ | $a_4$ | $a_5$ | $a_6$ | $a_7$ | $a_8$ | $a_9$ | $a_{10}$ | $a_{11}$ | $a_{12}$ | $a_{13}$ | $a_{14}$ | $a_{15}$ | $a_{16}$ | $a_{17}$ | $a_{18}$ |
|---|---|---|---|---|---|---|---|---|---|---|---|---|---|---|---|---|---|---|
| $a_1$ | 0 | 1 | 1 | 0 | 0 | 0 | 1 | 0 | 0 | 0 | 0 | 0 | 0 | 0 | 0 | 0 | 1 | 0 |
| $a_2$ | 0 | 0 | 1 | 0 | 0 | 0 | 1 | 0 | 0 | 0 | 0 | 0 | 0 | 0 | 0 | 0 | 0 | 0 |
| $a_3$ | 0 | 0 | 0 | 0 | 1 | 1 | 1 | 1 | 0 | 0 | 0 | 0 | 0 | 0 | 0 | 0 | 0 | 0 |
| $a_4$ | 0 | 1 | 1 | 0 | 0 | 0 | 0 | 0 | 0 | 0 | 0 | 0 | 0 | 0 | 0 | 0 | 0 | 0 |
| $a_5$ | 0 | 0 | 0 | 0 | 0 | 1 | 0 | 1 | 0 | 0 | 0 | 0 | 0 | 0 | 0 | 1 | 0 | 0 |
| $a_6$ | 0 | 0 | 0 | 0 | 0 | 0 | 0 | 0 | 0 | 0 | 0 | 0 | 0 | 0 | 0 | 0 | 1 | 0 |
| $a_7$ | 0 | 0 | 0 | 0 | 1 | 1 | 0 | 0 | 0 | 1 | 0 | 0 | 0 | 0 | 0 | 0 | 1 | 0 |
| $a_8$ | 0 | 0 | 0 | 0 | 0 | 1 | 0 | 0 | 0 | 1 | 0 | 0 | 0 | 0 | 0 | 0 | 0 | 0 |
| $a_9$ | 0 | 0 | 0 | 0 | 0 | 0 | 0 | 1 | 0 | 1 | 0 | 0 | 0 | 0 | 0 | 0 | 0 | 0 |
| $a_{10}$ | 0 | 0 | 0 | 0 | 0 | 1 | 0 | 0 | 0 | 0 | 0 | 0 | 0 | 0 | 0 | 0 | 0 | 0 |
| $a_{11}$ | 0 | 0 | 1 | 1 | 0 | 0 | 0 | 0 | 0 | 0 | 1 | 0 | 0 | 0 | 0 | 0 | 0 | 0 |
| $a_{12}$ | 0 | 1 | 0 | 1 | 0 | 0 | 0 | 0 | 0 | 0 | 0 | 0 | 0 | 0 | 0 | 0 | 0 | 0 |
| $a_{13}$ | 0 | 0 | 0 | 0 | 0 | 0 | 0 | 0 | 0 | 0 | 1 | 1 | 0 | 1 | 1 | 0 | 0 | 1 |
| $a_{14}$ | 0 | 1 | 1 | 0 | 0 | 0 | 0 | 0 | 0 | 1 | 0 | 0 | 0 | 0 | 0 | 1 | 0 | 0 |
| $a_{15}$ | 0 | 0 | 0 | 0 | 0 | 0 | 0 | 0 | 0 | 0 | 0 | 0 | 0 | 0 | 0 | 1 | 0 | 0 |
| $a_{16}$ | 0 | 0 | 0 | 0 | 0 | 1 | 0 | 0 | 0 | 0 | 0 | 0 | 0 | 0 | 0 | 0 | 1 | 0 |
| $a_{17}$ | 0 | 0 | 0 | 0 | 0 | 0 | 0 | 0 | 0 | 1 | 0 | 0 | 0 | 0 | 0 | 0 | 0 | 0 |
| $a_{18}$ | 1 | 1 | 0 | 0 | 0 | 0 | 0 | 0 | 0 | 0 | 0 | 0 | 0 | 0 | 0 | 0 | 0 | 0 |

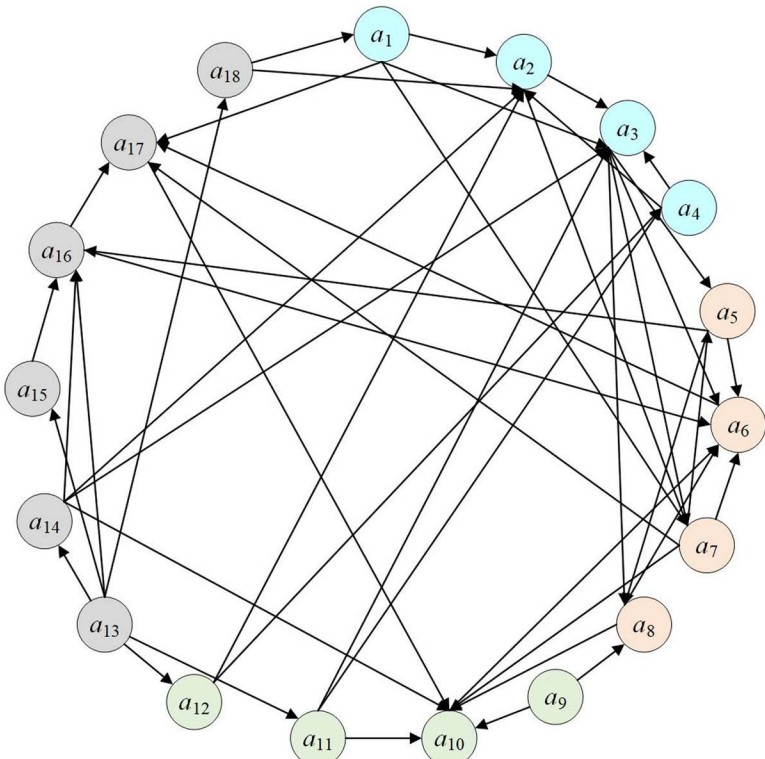

**Fig 8. Complex network diagram of railway safety risk factors.**

Table 7, the top five risk factors with the highest out-degree are $a_{13}$ (Incomplete Regulations and Procedures), $a_1$ (Inadequate Professional Skill Level), $a_3$ (Unsafe Behaviors), $a_7$ (Insufficient Maintenance and Repair), and $a_{14}$ (Inadequate Supervision and Management). These high out-degree factors are predominantly management and personnel factors, which are more likely to induce accidents. Conversely, the top five risk factors with the highest in-degree are $a_{10}$ (Foreign Object Intrusion), $a_6$ (Poor Equipment Performance or Malfunction), $a_3$ (Unsafe Behaviors), $a_7$ (Insufficient Maintenance and Repair), and $a_2$ (Insufficient Safety Awareness). These high in-degree factors are primarily equipment and personnel factors, which are more susceptible to being influenced by other risks.

Intermediary centrality emphasizes the number of shortest paths in the network that pass through a particular node. A node with higher intermediary centrality is a key factor in controlling risk transmission. The top five risk factors with the highest intermediary centrality are $a_7$ (inadequate maintenance), $a_3$ (unsafe behavior), $a_8$ (security protection failure), $a_{15}$ (insufficient coordination and organization ability), and $a_{10}$ (foreign body intrusion). By mitigating these risk factors, the connectivity of the risk network can be effectively disrupted, thereby preventing the formation of systemic risks.

**(2) Accessibility analysis of risk factors.** Reachability refers to the ease with which one vertex can be accessed from another in a network, reflecting the potential for risk factors to influence other factors. In this study, accessibility is defined as the ability of risk factors to trigger a chain reaction through coupling relationships. If there is at least one effective transmission path from factor $i$ to factor $j$ (such as $i \rightarrow k \rightarrow j$), then factor $i$ is said to be accessible to factor $j$. In conjunction with the *N-K* model, the reachability of the railway safety risk network has been thoroughly analyzed, and the reachability of 18 risk factors has been categorized into four types of risks. The specific results are presented in Table 8.

**(3) Model result correction.** The *N-K* model employs accident data analysis, which is characterized by strong objectivity and accuracy. However, its application is limited to the analysis of four major risk factors: personnel, equipment, environment, and management. In contrast, the construction of complex networks may involve certain subjective factors, and there is a possibility that the identified core risks might not necessarily correlate with a high probability of actual

**Table 7. Centrality of risk factors.**

| Risk factors | Closeness centrality | | Betweenness centrality/% |
|---|---|---|---|
| | Out-degree/% | In-degree/% | |
| $a_1$ | 8.33 | 2.78 | 2.21 |
| $a_2$ | 4.17 | 11.1 | 4.41 |
| $a_3$ | 8.33 | 13.51 | 11.03 |
| $a_4$ | 4.17 | 5.41 | 1.47 |
| $a_5$ | 6.25 | 5.41 | 6.25 |
| $a_6$ | 2.08 | 13.51 | 4.41 |
| $a_7$ | 8.33 | 13.16 | 17.6 |
| $a_8$ | 4.17 | 7.89 | 10.3 |
| $a_9$ | 4.17 | 0.00 | 0.00 |
| $a_{10}$ | 2.08 | 15.7 | 7.35 |
| $a_{11}$ | 6.25 | 2.63 | 2.94 |
| $a_{12}$ | 4.17 | 2.63 | 2.21 |
| $a_{13}$ | 12.5 | 0.00 | 0 |
| $a_{14}$ | 8.33 | 2.63 | 2.21 |
| $a_{15}$ | 2.08 | 2.63 | 10.0 |
| $a_{16}$ | 4.17 | 10.26 | 2.21 |
| $a_{17}$ | 2.08 | 10.0 | 1.47 |
| $a_{18}$ | 4.17 | 2.44 | 3.68 |

**Table 8. Accessibility analysis of risk factors.**

| Risk factors | Personnel | Equipment | Environment | Management | Potential coupling forms |
|---|---|---|---|---|---|
| $a_1$ | 1 | 1 | 0 | 1 | Personnel-Equipment- Management |
| $a_2$ | 1 | 1 | 0 | 0 | Personnel-Equipment |
| $a_3$ | 0 | 1 | 0 | 0 | Personnel-Equipment |
| $a_4$ | 1 | 0 | 0 | 0 | Personnel |
| $a_5$ | 1 | 1 | 1 | 1 | Personnel-Equipment-Environment-Management |
| $a_6$ | 1 | 0 | 0 | 1 | Personnel-Equipment- Management |
| $a_7$ | 1 | 1 | 1 | 1 | Personnel-Equipment-Environment-Management |
| $a_8$ | 0 | 1 | 1 | 0 | Equipment-Environment |
| $a_9$ | 0 | 1 | 1 | 0 | Equipment-Environment |
| $a_{10}$ | 0 | 1 | 1 | 0 | Equipment-Environment |
| $a_{11}$ | 1 | 0 | 1 | 0 | Personnel-Environment |
| $a_{12}$ | 1 | 0 | 1 | 0 | Personnel-Environment |
| $a_{13}$ | 1 | 0 | 1 | 1 | Personnel-Environment-Management |
| $a_{14}$ | 1 | 0 | 1 | 1 | Personnel-Environment-Management |
| $a_{15}$ | 0 | 0 | 0 | 1 | Management |
| $a_{16}$ | 0 | 1 | 0 | 1 | Equipment-Environment |
| $a_{17}$ | 0 | 0 | 1 | 1 | Environment-Management |
| $a_{18}$ | 1 | 0 | 0 | 1 | Personnel-Management |

accidents. To address these limitations, the *N-K* model was integrated into the complex network framework and combined with the reachability analysis presented in Table 7. Based on the N – K model, the coupling values T corresponding to different coupling forms are calculated. Within the complex network model, the potential risk chains among various risk factors are identified, the potential coupling forms of the corresponding risk factors are determined, and the closeness centrality and intermediary centrality of the risk nodes are modified by leveraging the potential risk chain coupling forms. The risk coupling values were then used to adjust the proximity and intermediary centrality of the risk nodes. The revised results are illustrated in Fig 9–11.

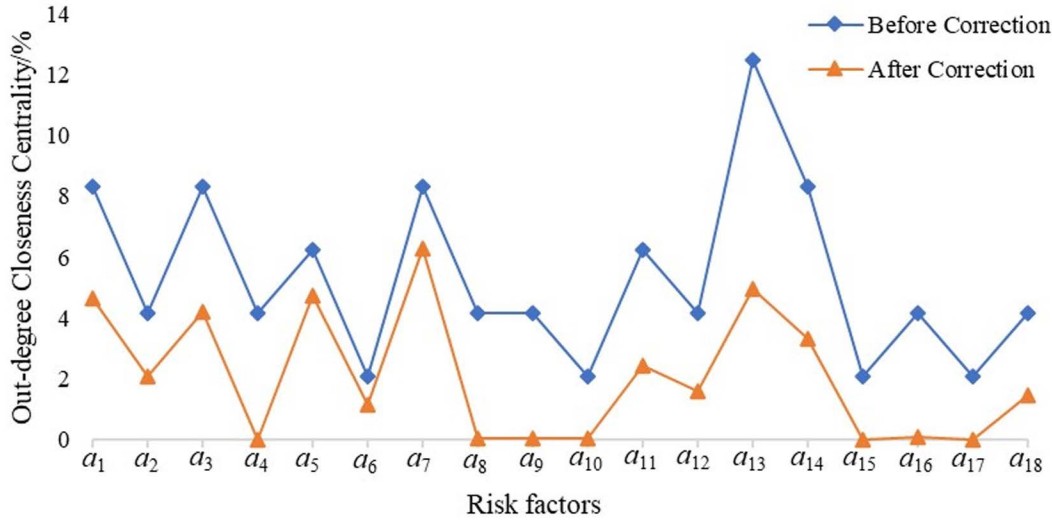

**Fig 9. Complex network centrality of railway safety risk (a)Out-degree Closeness Centrality.**

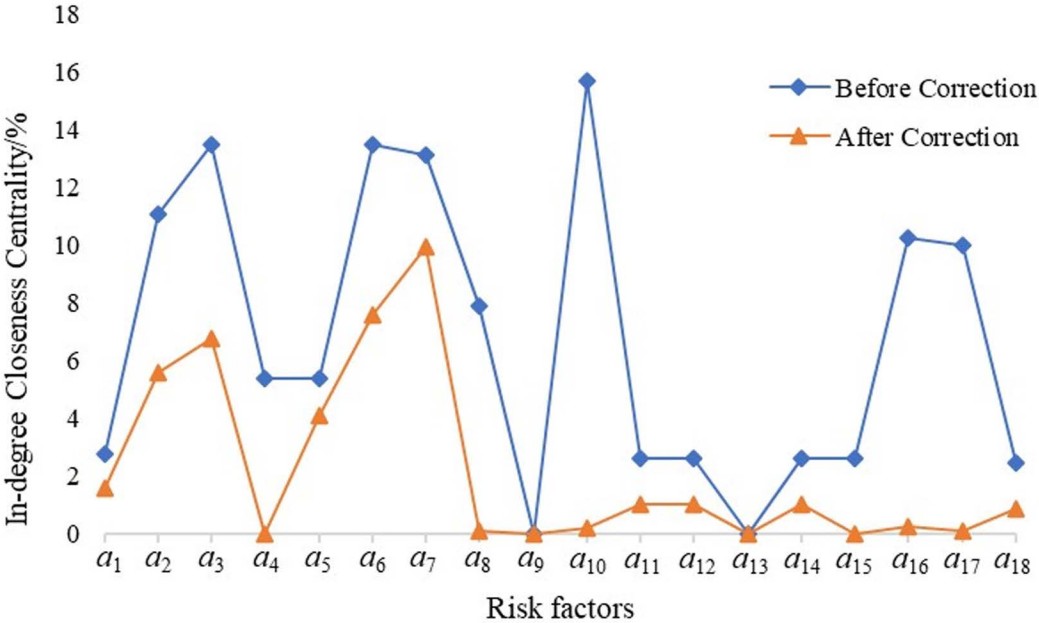

**Fig 10. Complex network centrality of railway safety risk (b)In-degree Closeness Centrality.**

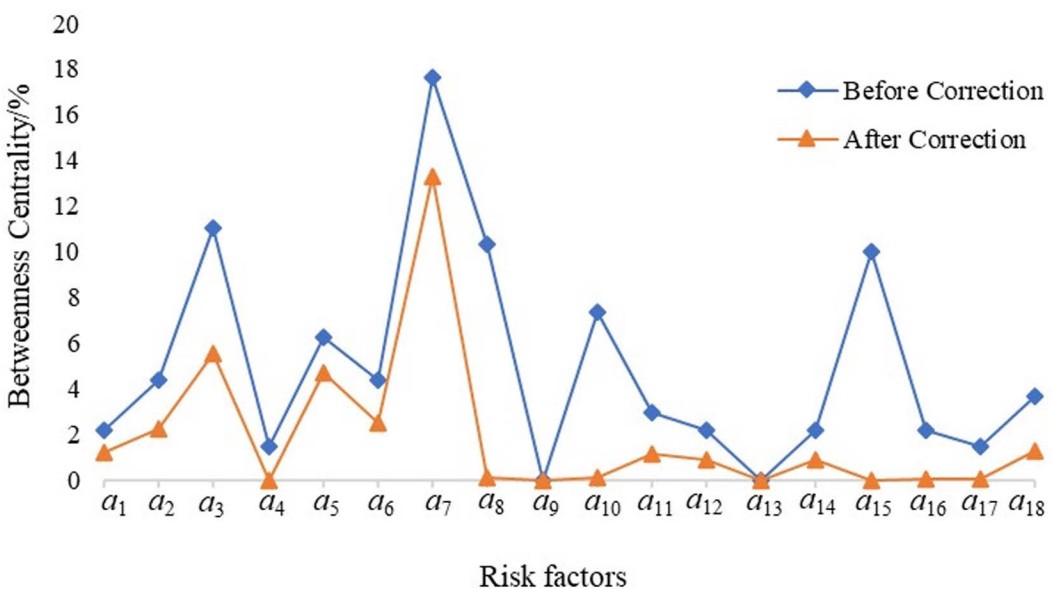

**Fig 11. Complex network centrality of railway safety risk (c)Betweenness Centrality.**

The key risk factors of railway safety risk are identified and sorted out according to Fig 9–11. As shown in Table 9, the factors $a_7$ (Insufficient Maintenance and Repair) and $a_3$ (Unsafe Behaviors) have consistently exhibited higher values both before and after correction. This indicates that these two factors are critical contributors to railway safety risk. Specifically,

**Table 9. Key factors of railway safety risk.**

| Complex network centrality measure | Before correction | After correction |
|---|---|---|
| Out-degree Closeness Centrality | $a_{13}$、$a_1$、$a_3$、$a_7$、$a_{14}$ | $a_7$、$a_{13}$、$a_1$、$a_5$、$a_3$ |
| In-degree Closeness Centrality | $a_7$、$a_3$、$a_8$、$a_{15}$、$a_{10}$ | $a_7$、$a_6$、$a_3$、$a_2$、$a_5$ |
| Betweenness Centrality | $a_7$、$a_3$、$a_8$、$a_{15}$、$a_{10}$ | $a_7$、$a_3$、$a_5$、$a_6$、$a_2$ |

inadequate maintenance and unsafe behavior play extremely critical roles within the entire railway safety risk system and have a significant impact on railway operational safety. Therefore, these factors must be closely monitored and controlled on an ongoing basis.

Factors that exhibit substantial changes after correction include $a_5$ (Missing Equipment Functionality), $a_6$ (Poor Equipment Performance or Malfunction), and $a_2$ (Insufficient Safety Awareness). This suggests that these factors possess a high degree of correctability. In the context of daily management and maintenance of railway system operations, implementing corresponding preventive measures—such as regular equipment inspections, updating aging equipment, and enhancing personnel safety training—can effectively reduce railway safety risks. These measures are likely to yield positive feedback and ensure the safe and stable operation of the railway system.

## Conclusion

This study proposes a coupling analysis method for railway safety risk based on the *N-K* model and complex network theory. It defines the coupling types of railway safety risk, which are composed of personnel, equipment, environment, and management factors. The coupling mechanism of railway safety risk is explained through the analysis and calculation of couplings between different types of risk factors. The complex network of railway safety risk is modified using coupling interaction information values, and key risk factors leading to railway accidents are identified. This approach enhances the accuracy of railway safety risk identification. The main conclusions are as follows:

(1) Compared to traditional static risk assessment methods, such as the tensor analysis and fault tree models referenced in prior studies, this paper explores risk coupling relationships using the *N-K* model. It examines the nonlinear amplification effects of multi-factor interactions and reveals that the coupling value of four factors surpasses that of a single factor, confirming a super-linear increase in risk with the number of coupling factors. Additionally, the research findings, particularly regarding personnel-equipment coupling, indicate that in dual-factor scenarios, the highest coupling value occurs between personnel and equipment. This corroborates that human-machine interaction is the primary cause of accidents and underscores the necessity of synchronized control to disrupt the accident chain.

(2) The centrality analysis of the complex network of railway safety risk factors, based on the N-K model, identifies inadequate maintenance and unsafe behaviors as key factors leading to railway accidents. Addressing these two risk factors at their source can effectively disrupt the risk network and reduce the incidence of railway accidents. Additionally, after correcting the coupling interaction values using the N-K model, three risk factors—equipment function loss, poor performance or failure of equipment, and lack of safety awareness—were found to have high centrality values. Therefore, relevant railway departments should prioritize safety education and training for employees to enhance their awareness of potential dangers. Furthermore, regular inspection, maintenance, and repair of equipment and facilities are crucial to ensure the safe and reliable operation of railway infrastructure and rolling stock.

(3) Through complex network analysis, this study overcomes the constraints of traditional accident path prediction methods and unveils the dynamic propagation characteristics of key risk nodes within railway systems. Inadequate maintenance and repair, exhibiting a high degree of intermediacy, has emerged as the central hub for risk diffusion. Controlling this factor can effectively disrupt risk propagation pathways. While unsafe behaviors serve as key triggers,

their low out-degree and high in-degree suggest that accidents typically require activation through equipment or managerial nodes. These findings not only empirically validate the human factors core theory highlighted in prior literature but also expose the equipment dependency inherent in human factor propagation, offering novel targets for precise prevention and control measures.

This study expands the theoretical framework for railway safety risk analysis, reinforcing the theoretical underpinnings of the dual prevention mechanism for railway safety accident risk. It provides a key theoretical reference and further enriches the theoretical and methodological framework for railway safety accident risk prevention and control. However, in actual railway operations, each risk factor evolves dynamically over time, and the coupling effect of risk varies significantly at different time points. In light of this, future research will focus on incorporating the influence of time parameters and will be dedicated to developing a dynamic analysis model for the coupling effect of railway operation safety risks. This effort aims to provide robust theoretical and methodological support for enhancing the overall safety level of the railway system.

## Author contributions

**Data curation:** Lin Zhao.

**Funding acquisition:** Zhan Guo.

**Investigation:** Jinghui Liu.

**Validation:** Gaolei Wang.

**Writing – original draft:** Jiaxu Chen.

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
