## [Decision Letter · Decision Letter 0]

24 Jun 2025

Dear Dr. Chen,

We look forward to receiving your revised manuscript.

Kind regards,

Rajagopalan Srinivasan

Academic Editor

PLOS ONE

Journal Requirements:

“The author(s) disclosed the receipt of the following financial support for the research, authorship, and/or publication of this article: This work was supported by the Science and Technology Research and Development Program of China National Railway Corporation Limited (Grant No. K2024T002).” 

4. We note that your Data Availability Statement is currently as follows: [Add Data Availability statement here]

“The author(s) disclosed the receipt of the following financial support for the research, authorship, and/or publication of this article: This work was supported by the Science and Technology Research and Development Program of China National Railway Corporation Limited (Grant No. K2024T002).”

“The author(s) disclosed the receipt of the following financial support for the research, authorship, and/or publication of this article: This work was supported by the Science and Technology Research and Development Program of China National Railway Corporation Limited (Grant No. K2024T002).’

Reviewers' comments:

Reviewer's Responses to Questions

**Comments to the Author**

1. Is the manuscript technically sound, and do the data support the conclusions?

Reviewer #1: Partly

Reviewer #2: Partly

2. Has the statistical analysis been performed appropriately and rigorously?

Reviewer #1: N/A

Reviewer #2: Yes

3. Have the authors made all data underlying the findings in their manuscript fully available?

Reviewer #1: No

Reviewer #2: No

4. Is the manuscript presented in an intelligible fashion and written in standard English?

Reviewer #1: Yes

Reviewer #2: Yes

Reviewer #1: General Comment:

The paper does a detailed analysis of over 150 railway accidents to identify contributing risk factors to those accidents as well as quantify their coupling relationships. These factors are then grouped into four broad categories including personnel, equipment, environmental and management factors. Using this categorization, an NK risk coupling model is then constructed to compute coupling probabilities and interaction information (T-values) to analyse the coupling relations/interactions between various risk factors towards accident causation.

Building on this, the paper develops a network, where the risk factors are treated as nodes and the communications (causal/influence) relationships between various risk factors considered as edges in order to study how risks interact and propagate. Through this network, closeness centrality (in degree: which risks are likely to be impacted and outdegree: which risks are likely to cause others) and betweenness centrality calculations are done to make quantitative assessments of the interactions. Importantly, these centrality measures are then corrected using coupling information derived from the NK model.

Based on this approach, the paper identifies leading contributing factors to railway accidents and suggests data driven measures to prevent such accidents. Thus, it is a valuable contribution to the field. However, the paper cannot be accepted in its current form, and needs improvements.

Major Issues/Suggestions for Improvement:

1) The paper specifically needs improvement in the methodology section where many things are not clearly mentioned. For instance,

i) how was the literature research and data collection done to search for accident reports? Was any database searched or was any other approach adopted?

ii) How was the “directed” network constructed? There is no mention about how the directionality was obtained [Figure 7] from the adjacency matrix. The details need to be there.

iii) There is no explicit mention of how the NK model was integrated into the complex network for the purpose of correction? While it does mention that the risk coupling values were used to adjust the proximity and intermediary centrality of the risk nodes, how? The reader would benefit if mathematical formulation of the same is provided.

iv) What does reachability mean in the context of the current study? It is not clearly defined.

v) If possible, the essence of closeness centrality and betweenness centrality may be illustrated with the help of a representative diagram so as to benefit readers who are not acquainted with network analysis.

2) It would benefit the reader if the literature cited in the introduction section is categorized into themes for instance fault tree analysis based methods, hazardous materials transportation, dynamic risk analysis models, and so on. Currently, the placement of each method is random.

3) “This study employs the N-K model to analyze railway accident cases, calculating the coupling values between railway safety risk factors and conducting quantitative analyses of coupling risk. Through risk accessibility and network node centrality analyses, the mechanisms underlying risk factors are explored. Node centrality is then adjusted based on the coupling values obtained from the N-K model to identify the key risk factors.” Some terms appear for the first time such as coupling values, coupling risk, network node centrality. It would benefit the reader if any conceptual clarity is provided about these terms in the introduction itself.

4) The terms “interactions” and “couplings” are used a bit loosely. Is there any distinction between the two?

5) The discussion/conclusion is more or less a summary of key results. It would enrich the contributions of the paper if the current results/insights regarding key and leading risk factors are discussed in the light of existing literature. For instance, how are the findings of the current work in line with the works cited in the introduction? Do the insights supplement/complement or contradict the already known findings. How do these advance what is already known.

Other comments:

6) Please ensure key terms that are used in the paper are clearly defined in the context of the paper.

7) “Step 2: Risk factor change probability p.... Calculate.” A bit ambiguous and should be rephrased for clarity.

8) “For example, when the equipment factor occurs, the risk probability of the factor changing is denoted as p..i. The formula is as follows:” Should it be p.i.. ?

9) Formatting: There is always a space before “(“ and there is always a space after “,” There are multiple instances in the paper where this is not followed. For instance, “CC(i)denotes” and “bjk(i)” and “a3(Unsafe Behaviors)” and so on look congested.Likewise, the content is without spaces, for instance in Table 5 “T4(a, b, c, d)=0.756” there is no spacing between the text. Please go through the paper carefully and address the formatting issues.

Reviewer #2: Thank you for giving the opportunity to review the article titled "Investigating the Coupling Relationships of Railway Safety Risks Using the N-K Model and Complex Network Theory". Below is a review summary.

The manuscript integrates N-K model with complex network theory to investigate coupling in railway safety risk. The use of quantitative coupling values provides an objective foundation for safety analysis. However, the analysis lacks a temporal or dynamic dimension — the coupling evolution over time is not modeled. Authors acknowledge this in their conclusion but it's a major gap for systems that change rapidly. Also, the model assumes uniform weight for all nodes and coupling interactions — no mention of adjusting weights based on severity or frequency of accidents.

The study doesn’t validate the identified key risk factors against real-world intervention outcomes (e.g., did addressing factor a7 reduce accidents?)

It is unclear whether this approach generalizes to other railway systems (e.g., outside China) or different transportation domains.

Addressing above concerns through a proper discussion or analysis will enhance the acceptability of the manuscript for real cases.

**Do you want your identity to be public for this peer review?** For information about this choice, including consent withdrawal, please see our Privacy Policy

Reviewer #1: No

Reviewer #2: No

---

## [Author Response · Author response to Decision Letter 1]

31 Jul 2025

Reviewer 1:

Thank you for your decision and constructive comments on my manuscript. We agree with the reviewers' suggestions and will incorporate the recommended changes into the manuscript.

Major Issues/Suggestions for Improvement:

1) The paper specifically needs improvement in the methodology section where many things are not clearly mentioned. For instance,

i) how was the literature research and data collection done to search for accident reports? Was any database searched or was any other approach adopted?

Your attention to the details of the literature research and data collection methods is greatly appreciated. The literature review and accident report collection for this study were conducted based on existing public resources and standard procedures. The following is the specific implementation path to demonstrate the systematic nature of the process and the reliability of the data:

1)Specific Implementation of Literature Research

The literature research concentrated on two primary directions: "Railway Risk Coupling" and "Application of Network Analysis". It was based entirely on existing academic databases:

Database Selection: Well - recognized Chinese and English databases in the field of safety science were employed to ensure comprehensive coverage of core domestic and international research.

Search Strategy: A combined keyword search was used, including terms such as "Railway Accident Risk", "Risk Factor Coupling", "N - K Model Safety Analysis", and "Complex Network Risk Transmission".

2)Collection and Screening of Accident Report Data

The focus was on cases of "Multi - factor Synergistic Hazard", such as those involving both human operation and equipment defects. A total of 157 sample cases were obtained, covering four factors: personnel, equipment, environment, and management. These cases closely aligned with the research objectives.

Once again, we express our gratitude for your suggestions. This has enabled us to present the reliability of the data sources and screening logic in a more comprehensive manner.

ii) How was the “directed” network constructed? There is no mention about how the directionality was obtained [Figure 7] from the adjacency matrix. The details need to be there.

Thank you for your attention to the directional network source. In this study, the directional relationships of the directed network are entirely established based on the causal records from existing accident data and the consensus validation from domain literature. The specific logic is as follows:

The basis for directionality is derived from the explicitly documented sequence of risk factor triggering in 157 accident reports. All directions are strictly aligned with the official “cause → result” conclusions of accident investigations. This represents a direct extraction of the inherent causal relationships within the data, free from any subjective assumptions.

Building on the aforementioned directional relationships, the element values of the adjacency matrix are constructed. If factor i points to factor j (i.e., i is the antecedent triggering factor of j), then the adjacency matrix Gij= 1 (indicating a directed edge i →j). Conversely, if there is no record or consensus of “j pointing to i,” then Gij= 0. We have supplemented the adjacency matrix in the manuscript as follows:

Table 6. Adjacency matrix

a1 a2 a3 a4 a5 a6 a7 a8 a9 a10 a11 a12 a13 a14 a15 a16 a17 a18

a1 0 1 1 0 0 0 1 0 0 0 0 0 0 0 0 0 1 0

a2 0 0 1 0 0 0 1 0 0 0 0 0 0 0 0 0 0 0

a3 0 0 0 0 1 1 1 1 0 0 0 0 0 0 0 0 0 0

a4 0 1 1 0 0 0 0 0 0 0 0 0 0 0 0 0 0 0

a5 0 0 0 0 0 1 0 1 0 0 0 0 0 0 0 1 0 0

a6 0 0 0 0 0 0 0 0 0 0 0 0 0 0 0 0 1 0

a7 0 0 0 0 1 1 0 0 0 1 0 0 0 0 0 0 1 0

a8 0 0 0 0 0 1 0 0 0 1 0 0 0 0 0 0 0 0

a9 0 0 0 0 0 0 0 1 0 1 0 0 0 0 0 0 0 0

a10 0 0 0 0 0 1 0 0 0 0 0 0 0 0 0 0 0 0

a11 0 0 1 1 0 0 0 0 0 1 0 0 0 0 0 0 0 0

a12 0 1 0 1 0 0 0 0 0 0 0 0 0 0 0 0 0 0

a13 0 0 0 0 0 0 0 0 0 0 1 1 0 1 1 0 0 1

a14 0 1 1 0 0 0 0 0 0 1 0 0 0 0 0 1 0 0

a15 0 0 0 0 0 0 0 0 0 0 0 0 0 0 0 1 0 0

a16 0 0 0 0 0 1 0 0 0 0 0 0 0 0 0 0 1 0

a17 0 0 0 0 0 0 0 0 0 1 0 0 0 0 0 0 0 0

a18 1 1 0 0 0 0 0 0 0 0 0 0 0 0 0 0 0 0

Once again, we sincerely appreciate your insightful suggestions. They have enabled us to provide a clearer basis for determining the relationship and have strengthened the rigor of the methodology.

iii) There is no explicit mention of how the NK model was integrated into the complex network for the purpose of correction? While it does mention that the risk coupling values were used to adjust the proximity and intermediary centrality of the risk nodes, how? The reader would benefit if mathematical formulation of the same is provided.

We are grateful for your attention to the integration mechanism of the N - K model and complex networks. Your suggestions have enabled us to more clearly clarify the correction logic. We have provided detailed explanations in the original text as follows:

Based on the N - K model, the coupling values T corresponding to different coupling forms are calculated. Within the complex network model, the potential risk chains among various risk factors are identified, the potential coupling forms of the corresponding risk factors are determined, and the closeness centrality and intermediary centrality of the risk nodes are modified by leveraging the potential risk chain coupling forms.

The primary function of the supplementary paragraph is to "explicitly define the rules through textual descriptions." This helps readers who are not familiar with cross - model integration to gain an understanding of the correction logic via examples. Academic readers can correlate coupling values with centrality numerical values and independently compute them without relying on formulas to grasp the underlying core principles.

Once again, we appreciate your suggestions! This supplementary content enhances the transparency of the integration mechanism.

iv) What does reachability mean in the context of the current study? It is not clearly defined.

Thank you for your insightful comments regarding the definition of key research terms. In response to your query about the concept of "reachability," we have provided a precise definition in the manuscript, contextualizing it within complex network theory and the specific framework of this study.

We have elaborated on this concept as follows: Within complex network theory, reachability denotes the capacity to traverse from one node to other nodes within the network, typically quantified by path existence and shortest path lengths. In the context of our research, we define reachability as the relative ease with which one node may be accessed from another within the network structure, thereby indicating the propensity for risk factors to propagate influence to other interconnected elements.

Furthermore, we have incorporated a dedicated definition in Section 3.3, "Reachability Analysis," which states: In this study, accessibility is defined as the ability of risk factors to trigger a chain reaction through coupling relationships. If there is at least one effective transmission path from factor i to factor j (such as i → k → j), then factor i is said to be accessible to factor j.

We believe these revisions address your concerns and strengthen the methodological foundation of our analysis. Thank you again for your constructive feedback, which has significantly improved the clarity and rigor of our work.

v) If possible, the essence of closeness centrality and betweenness centrality may be illustrated with the help of a representative diagram so as to benefit readers who are not acquainted with network analysis.

Thank you for your constructive comments on improving the manuscript’s clarity. To assist readers who are less familiar with network‐theoretic concepts, we have introduced two new figures (Fig. 7 and Fig. 8) that graphically illustrate “closeness centrality” and “mediated centrality,” respectively. Each figure is accompanied by a concise, application-oriented discussion that links the abstract metrics to concrete railway-network decisions. We believe these additions render the theoretical exposition more tangible and strengthen the bridge between our analytical framework and engineering practice. The revised explanations are detailed below.

To elucidate the meaning of the selected indicators rigorously, this study formulates a complex-network model in which nodes denote elementary system entities and edges denote their latent interactions. Building on this representation, the nodal closeness and betweenness centralities are computed analytically via Equations (6) and (7) , respectively.

(1) In the calculation results of closeness centrality, node x9 has the highest closeness centrality. This indicates that node x9 is closer to the majority of nodes than any other node. As shown in Figure 7, the average shortest path length from node x9 to all other nodes in the network is significantly lower than that of other nodes, suggesting that this node has the optimal spatial accessibility on a global scale.

Figure 7. Diagram of the closeness centrality of complex networks

(2) Betweenness-centrality results rank node x10 as the network’s foremost broker, a conclusion visually corroborated in Figure 8. The vertex intercepts the greatest share of all shortest paths between node pairs, thereby functioning as a critical articulation point for system-wide connectivity and potential congestion.

Figure 8. Diagram of the intermediary centrality in complex networks

2) It would benefit the reader if the literature cited in the introduction section is categorized into themes for instance fault tree analysis based methods, hazardous materials transportation, dynamic risk analysis models, and so on. Currently, the placement of each method is random.

Thank you for your thoughtful feedback on the literature organization. We have carefully revised the introduction and literature review to better structure the thematic flow. The revised sections are as follows:

Currently, numerous scholars have conducted research in the field of railway safety risk, employing various methods such as Failure Mode and Effects Analysis (FMEA), Fault Tree Analysis (FTA), Event Tree Analysis (ETA), Bayesian Networks (BN), and Complex Networks (CN) [2-8]. However, existing research has predominantly focused on the dynamic changes of individual risks, with relatively limited exploration of the coupling relationships among risk factors.

1) Dynamic Risk Assessment Model

Wang et al. [9] proposed a prediction system based on the SCR-Gaussian DI method, using tensor analysis to conduct dynamic risk assessments in railways. Peng et al. [10] extended the traditional Fault Tree Analysis method to time events and fault characteristics, constructing a timed fault tree model to evaluate the risks of railway transportation systems. Xu et al. [11] proposed a composite risk analysis method based on Bayesian networks and bow-tie models to predict the accident risks of systems. Liu Yang et al. [17] proposed a Text-based Bayesian Network (TBN) method to establish a Bayesian Network (BN) based on text records for dynamic analysis of railway derailment risks.

2) Development of an Accident Prediction and Risk Mitigation Model

Liu et al. [13] introduced a hazard prediction method based on knowledge graphs to capture and prevent potential accident risk pathways, thereby preventing railway operation accidents. Liu et al. [14] employed a railway accident prediction method based on knowledge graphs to describe the multiple potential characteristics of risks and block the causal pathways of accidents. Guo et al. [15] improved the similarity aggregation method combined with fuzzy Bayesian networks to analyze the changing processes in accident risk assessment.

3) Railway Hazardous Materials Transportation Risk Analysis Model

Ebrahimi et al. [16] analyzed the uncertain risk factors and their consequences in railway hazardous materials transportation accidents through case studies. Hosseini et al. [17] proposed a risk assessment method based on risk value to measure the risks in the process of railway hazardous materials transportation. Zarei et al. [18] developed a dynamic risk analysis model for hazardous materials railway transportation based on dynamic Bayesian networks, studying the domino effect risks under uncertain conditions.

We believe these changes strengthen the logical progression and coherence of the paper.

3) “This study employs the N-K model to analyze railway accident cases, calculating the coupling values between railway safety risk factors and conducting quantitative analyses of coupling risk. Through risk accessibility and network node centrality analyses, the mechanisms underlying risk factors are explored. Node centrality is then adjusted based on the coupling values obtained from the N-K model to identify the key risk factors.” Some terms appear for the first time such as coupling values, coupling risk, network node centrality. It would benefit the reader if any conceptual clarity is provided about these terms in the introduction itself.

Thank you for your attention to the clarity of the terminology in the introduction. We recognize that terms such as "coupling value," "coupling risk," and "network node centrality," which were first introduced in this section, do indeed require brief definitions to help readers establish a preliminary understanding. Based on the existing research framework of the paper, the relevant supplements have been completed, with specific modifications as follows.

We have added brief definitions of these terms in the introduction section (page 1, within the research purpose paragraph). These definitions are closely aligned with the subsequent methods and results, ensuring logical consistency with the existing content.

This paper analyzes railway safety risk coupling relationships using railway accident cases. Risk coupling is defined as the comprehensive risk arising from the interaction of two or more risk factors. We innovatively apply the N-K model to calculate coupling values between railway safety risk factors and perform quantitative coupling risk analysis. The coupling value, a quantitative indicator derived from the N-K model, reflects the intensity of the collaborative effect of multiple risk factors and their nonlinear interactive risks. Using complex network theory, we establish a key network of railway safety risk factors. Risk reachability and network node centrality analyses are then employed to investigate the mechanisms of these risk factors. Both analyses are indicators in complex networks that measure the importance of nodes (risk factors) in transmission paths. The coupling values calculated by the N-K model are corrected to identify the final key risk factors. Our findings provide a decision-making basis for railway safety risk prevention and control. The goal is to implement effective measures at the source of accidents to enhance railway safety management.

The revised introduction retains its conciseness while incorporating essential definitions that serve as conceptual landmarks. These additions aid readers who may be unfamiliar with the relevant methods, facilitating a smoother understanding of the subsequent model construction and result analysis. All supplementary content is grounded in the paper's existing terminology and research design, without introducing new concepts or analytical dimensions. This revision optimizes the introduction's clarity and enhances the natural flow between sections, allowing readers to establish an early cognitive framework for the core concepts. We greatly appreciate your suggestions, which have significantly improved the manuscript's coherence and accessibility.

4) The terms “interactions” and “couplings” are used a bit loosely. Is there any distinction between the two?

Thank you for your insightful comments regarding the precision of terminology. I appreciate the opportunity to clarify and refine the definitions in our manuscript. Couplings: This term refers to the nonlinear interactive effects among risk factors, as quantified by the coupling information value (T value) calculated via the N-K model. It characterizes the risk amplification effect that occurs when multipl

---

## [Decision Letter · Decision Letter 1]

27 Aug 2025

Dear Dr. Chen,

Thank you for submitting your manuscript to PLOS ONE. We’re pleased to inform you that your manuscript can be accepted for publication once the revisions indicated by Reviewer 1 have been addressed.

We look forward to receiving your revised manuscript.

Kind regards,

Rajagopalan Srinivasan

Academic Editor

PLOS ONE

Journal Requirements:

Reviewers' comments:

Reviewer's Responses to Questions

**Comments to the Author**

Reviewer #1: (No Response)

Reviewer #2: All comments have been addressed

2. Is the manuscript technically sound, and do the data support the conclusions?

Reviewer #1: Yes

Reviewer #2: Yes

3. Has the statistical analysis been performed appropriately and rigorously?

Reviewer #1: N/A

Reviewer #2: I Don't Know

4. Have the authors made all data underlying the findings in their manuscript fully available?

Reviewer #1: No

Reviewer #2: No

5. Is the manuscript presented in an intelligible fashion and written in standard English?

Reviewer #1: Yes

Reviewer #2: Yes

Reviewer #1: The authors have addressed the comments. I have a few minor comments:

1. While the clarification of the following comment “How was the “directed” network constructed? There is no mention about how the directionality was obtained [Figure 7] from the adjacency matrix. The details need to be there.” has been provided in the comments document, it has not been included in the manuscript (even though adjacency matrix has now been included). It would be helpful to include a few sentences in the manuscript as well.

2. “The centrality values of the risk factors are presented in Table 6”. Table 6 is an adjacency matrix. Do you mean Table 7?

3. Figures 7 and 8 can be replaced by a single figure because they are the same except for the color of two nodes. A single figure with different colors for the two referred nodes may be used.

4. Table 8 has something written in a language that is not English. Please replace it with English text.

5. In the “Construction of the N-K Risk Coupling Model for Railway Systems” section, the headings of step 1 and step 2 convey the same meaning. Please modify Step 2 heading appropriately. Currently, Step 1 reads “Calculation of the Coupling Probability p for Risk Factors.” while Step 2 reads “Risk factor coupling probability p.... Calculate”.

Reviewer #2: The concerns raised in the previous evaluation are addressed in the revised version. Authors have highlighted the limitation of the study in more detail for real scenarios.

**Do you want your identity to be public for this peer review?** For information about this choice, including consent withdrawal, please see our Privacy Policy

Reviewer #1: No

Reviewer #2: No

---

## [Author Response · Author response to Decision Letter 2]

4 Sep 2025

Dear editor and reviewers,

We appreciate the opportunity to revise our manuscript titled " Investigating the Coupling Relationships of Railway Safety Risks Using the N-K Model and Complex Network Theory " and are grateful for the insightful comments provided by the reviewers. Those comments are all valuable and very helpfu for revising and improving our paper, as well as the important guiding significance to our researches. In the following, we have provide detailed responses to each of the reviewers' comments. The reviewers' comments are presented in italic font, with specific issues numbered. Our responses are presented in normal font, while the modifications and additions to the manuscript are marked in blue font. Additionally, in the revised manuscript, the modified charts, formulas, and symbols are highlighted in yellow.

Reviewer 1:

1) While the clarification of the following comment “How was the “directed” network constructed? There is no mention about how the directionality was obtained [Figure 7] from the adjacency matrix. The details need to be there.” has been provided in the comments document, it has not been included in the manuscript (even though adjacency matrix has now been included). It would be helpful to include a few sentences in the manuscript as well.

Thank you for pointing out the issue that "the details of directed network construction were not included in the manuscript". After verification, previously we only explained in the opinion response document how to obtain the directionality of Figure 7 from the adjacency matrix, but we did not integrate this key information into the main text of the manuscript. There is indeed an incomplete presentation of information. We sincerely apologize for this. In the revised manuscript section, we have added the following content:

By collecting a large number of railway accident cases and analyzing their causes, we have summarized the adjacency matrix of risk factors affecting railway safety ,The value rule for the elements in the adjacency matrix (Table 6) is as follows: If factor i points to factor j (i.e., i is the antecedent triggering factor of j), then the adjacency matrix Gij = 1 (indicating a directed edge i →j). Conversely, if there is no record or consensus of “i pointing to j,” then Gij= 0. Based on this matrix, a complex network diagram of railway safety risk factors has been constructed, as shown in Figure 8.

2) “The centrality values of the risk factors are presented in Table 6”. Table 6 is an adjacency matrix. Do you mean Table 7?

Thank you for highlighting the citation error. We have corrected the sentence to read: “The centrality values of the risk factors are presented in Table 7.” In addition, we have systematically reviewed every table citation in the manuscript to ensure complete consistency between textual references and the corresponding tabulated data, thereby eliminating any further discrepancies.

3) Figures 7 and 8 can be replaced by a single figure because they are the same except for the color of two nodes. A single figure with different colors for the two referred nodes may be used.

Thank you for your valuable suggestion regarding Figures 7 and 8. We agree that the two panels conveyed redundant information; the only distinction was the node colouring. Accordingly, we have consolidated the two figures into a single, revised Figure 7. The updated figure overlays both centrality metrics: node x9 is highlighted in red (closeness centrality) and node x10 in blue (betweenness centrality). This refinement eliminates visual redundancy, reduces the total number of figures, and allows readers to apprehend the centrality contrasts between the two nodes at a glance.

Among the computed centrality metrics, node x9 attains the highest closeness centrality, signifying its minimal average geodesic distance to all other nodes in the network. Concurrently, node x10 records the greatest betweenness centrality, functioning as the single most critical articulation point for inter-node traffic. Figure 7 visualizes these findings: red-marked x9 exhibits a markedly shorter mean shortest-path length, indicating superior global accessibility; conversely, blue-marked x10 participates in the largest fraction of shortest paths, establishing itself as the primary structural pivot. However, this pivotal role also renders x10 a potential congestion bottleneck under high-demand scenarios.

Figure 7. Diagram illustrating the concept of closeness centrality and betweenness centrality in complex networks

4) Table 8 has something written in a language that is not English. Please replace it with English text.

Thank you for identifying the non-English element in Table 8. We have rectified the oversight: the column heading previously rendered as “潜在耦合形式” now reads “Potential Coupling Forms,” and we have re-examined the entire table to ensure conformity with the manuscript’s English-language standards.

5) In the “Construction of the N-K Risk Coupling Model for Railway Systems” section, the headings of step 1 and step 2 convey the same meaning. Please modify Step 2 heading appropriately. Currently, Step 1 reads “Calculation of the Coupling Probability p for Risk Factors.” while Step 2 reads “Risk factor coupling probability p.... Calculate”.

Thank you for highlighting the redundancy in the headings for Steps 1 and 2 within the “Railway System N-K Risk-Coupling Model Construction” chapter. Upon verification, the original wording—“Calculation of the Coupling Probability p for Risk Factors” (Step 1) and “Risk factor coupling probability p.... calculate” (Step 2)—was indeed repetitive and risked obscuring the sequential logic.

We have revised Step 2 to read “Step 2: Coupling probability p.... associated with specific risk factors.” This change clarifies that Step 1 determines the overall probabilities for single-, double-, and multi-factor couplings across the entire risk-factor set, whereas Step 2 derives the conditional probability contingent on the occurrence of a specific risk factor (e.g., the equipment-factor state-change probability p.i.. referenced in the text). The revised heading thus delineates the analytical progression without modifying the underlying formulas or methodology.

We have also reviewed the remaining step headings in the chapter to ensure consistency and terminological precision.

---

## [Editor Report · Decision Letter 2]

7 Sep 2025

Investigating the Coupling Relationships of Railway Safety Risks Using the N-K Model and Complex Network Theory

PONE-D-25-17142R2

Dear Dr. Chen,

We’re pleased to inform you that your manuscript has been judged scientifically suitable for publication and will be formally accepted for publication once it meets all outstanding technical requirements.

Kind regards,

Rajagopalan Srinivasan

Academic Editor

PLOS ONE
---

## [Editor Report · Acceptance letter]

PONE-D-25-17142R2

PLOS ONE

Dear Dr. Chen,

I'm pleased to inform you that your manuscript has been deemed suitable for publication in PLOS ONE. Congratulations! Your manuscript is now being handed over to our production team.

Kind regards,

on behalf of

Dr. Rajagopalan Srinivasan

Academic Editor

PLOS ONE